# Variable susceptibility of intestinal organoid–derived monolayers to SARS-CoV-2 infection

**Kyung Ku Jang**[1], **Maria E. Kaczmarek**[2¤], **Simone Dallari**[1], **Ying-Han Chen**[1], **Takuya Tada**[2], **Jordan Axelrad**[3], **Nathaniel R. Landau**[2], **Kenneth A. Stapleford**[2]\*, **Ken Cadwell**[1,2,3]\*

**1** Kimmel Center for Biology and Medicine at the Skirball Institute, New York University Grossman School of Medicine, New York, New York, United States of America, **2** Department of Microbiology, New York University Grossman School of Medicine, New York, New York, United States of America, **3** Division of Gastroenterology and Hepatology, Department of Medicine, New York University Grossman School of Medicine, New York, New York, United States of America

¤ Current address: EcoHealth Alliance in New York, New York, United States of America
\* kenneth.stapleford@nyulangone.org (KAS); ken.cadwell@med.nyu.edu (KC)

**Data Availability Statement:** Gene expression raw data are available from the GEO with accession number GSE179949. All other relevant data are

## Abstract

Gastrointestinal effects associated with Coronavirus Disease 2019 (COVID-19) are highly variable for reasons that are not understood. In this study, we used intestinal organoid–derived cultures differentiated from primary human specimens as a model to examine inter-individual variability. Infection of intestinal organoids derived from different donors with Severe Acute Respiratory Syndrome Coronavirus 2 (SARS-CoV-2) resulted in orders of magnitude differences in virus replication in small intestinal and colonic organoid–derived monolayers. Susceptibility to infection correlated with angiotensin I converting enzyme 2 (*ACE2*) expression level and was independent of donor demographic or clinical features. *ACE2* transcript levels in cell culture matched the amount of ACE2 in primary tissue, indicating that this feature of the intestinal epithelium is retained in the organoids. Longitudinal transcriptomics of organoid-derived monolayers identified a delayed yet robust interferon signature, the magnitude of which corresponded to the degree of SARS-CoV-2 infection. Interestingly, virus with the Omicron variant spike (S) protein infected the organoids with the highest infectivity, suggesting increased tropism of the virus for intestinal tissue. These results suggest that heterogeneity in SARS-CoV-2 replication in intestinal tissues results from differences in ACE2 levels, which may underlie variable patient outcomes.

## Introduction

Intestinal organoid cultures have transformed our ability to investigate properties of the human intestinal epithelium. Consisting of organized epithelial cell clusters differentiated from somatic stem cells, intestinal organoids generated from endoscopic pinch biopsies are capable of self-renewal and recreate many of the structural, functional, and molecular characteristics of the tissue of origin [1]. Investigators have exploited these versatile properties of intestinal organoids to study infectious agents that are otherwise difficult to examine,

within the paper and its Supporting Information files S1 Data and S1 Raw images.

**Funding:** This work was funded by National Health Institute grants DK093668 (K.C.), AI121244 (K.C.), HL123340 (K.C.), AI130945 (K.C.), AI140754 (K.C.), DK124336 (K.C.), DA046100 (N.R.L), AI122390 (N.R.L), AI120898 (N.R.L), and K23DK124570 (J.A.) (https://www.nih.gov/grants-funding); a New York University Clinical & Translational Science Awards grant UL1TR001445 (K.C.) (https://med.nyu.edu/departments-institutes/clinical-translational-science/education/career-development-programs/ctsa-scholars-program); a Faculty Scholar grant from the Howard Hughes Medical Institute (K.C.) (https://www.hhmi.org); Crohn's & Colitis Foundation (K.C. and J.A.) (https://www.crohnscolitisfoundation.org); Kenneth Rainin Foundation (K.C.) (https://krfoundation.org); Judith & Stewart Colton Center of Autoimmunity (K.C. and J.A.) (https://med.nyu.edu/departments-institutes/medicine/research/colton-center-autoimmunity) and Public Health Service Institutional Research Training Award T32 AI7647 (M.E.K.) (https://researchtraining.nih.gov/programs/training-grants/t32). The funders had no role in study design, data collection and analysis, decision to publish, or preparation of the manuscript.

**Competing interests:** I have read the journal's policy and the authors of this manuscript have the following competing interests: K.C. has received research support from Pfizer, Takeda, Pacific Biosciences, Genentech, and Abbvie. K.C. has consulted for or received an honoraria from Puretech Health, Genentech, and Abbvie. K.C. holds U.S. patent 10,722,600 and provisional patent 62/935,035 and 63/157,225. K.C. is a co-investigator on the Post-Acute Sequelae of SARS-CoV-2 Infection Initiative funded by the NIH (OT2HL161847). J.A. has received research support from BioFire Diagnostics. J.A. reports consultancy fees, honorarium, or advisory board fees from BioFire Diagnostics, Janssen, Abbvie, and Pfizer. J.A. holds U.S. patent 2012/0052124A1.

**Abbreviations:** ACE2, angiotensin I converting enzyme 2; BSA, bovine serum albumin; COVID-19, Coronavirus Disease 2019; DEG, differentially expressed gene; DMEM, Dulbecco's Modified Eagle Medium; FBS, fetal bovine serum; IBD, inflammatory bowel disease; IFNβ, interferon beta; IFNλ2, interferon lambda 2; IPA, ingenuity pathway analysis; ISG, interferon-stimulated gene; MOI, multiplicity of infection; MTT, thiazolyl blue tetrazolium bromide; NP, nucleoprotein; NYU, New York University; OGM, Organoid Growth Medium; PCA, principal component analysis; PFU, plaque-

including viruses such as noroviruses [2–4]. Intestinal organoids can also inform our understanding of interindividual differences in disease susceptibility, such as elucidating the mechanisms by which mutations accumulate in patients with colorectal cancer [5,6]. Additionally, we and others have documented substantial heterogeneity in the growth, morphology, viability, and susceptibility to cytokine toxicity of human intestinal organoid lines [1,6–8]. However, how this heterogeneity relates to resistance of the intestinal epithelium to infectious agents remains unclear.

Although Severe Acute Respiratory Syndrome Coronavirus 2 (SARS-CoV-2) infection is primarily associated with dysfunction of the respiratory system, the gastrointestinal tract is also an established target organ in patients with Coronavirus Disease 2019 (COVID-19). As many as 60% of patients present with diarrhea, vomiting, abdominal pain, anorexia, and/or nausea [9–16]. Also, SARS-CoV-2 antigen in intestinal biopsies and viral RNA in the stool are readily detected, even after the virus is undetectable in respiratory samples [17–27]. To a limited extent, virions and infectious particles have been detected in patient intestinal and stool specimens [17,25,28,29]. Although the pathophysiological significance of these observations has not been resolved, the prolonged presence of viral antigens in the gut is likely to impact antibody evolution [17]. Consistent with a potential intestinal tropism, intestinal epithelial cells display robust expression of the SARS-CoV-2 receptor angiotensin I converting enzyme 2 (ACE2) and transmembrane proteases TMPRSS2 and TMPRSS4 that facilitate viral entry [30–32]. Further, human intestinal organoids derived from either somatic stem cells or inducible pluripotent stem cells support SARS-CoV-2 reproduction [32–35]. These studies have shown that *ACE2* expression levels can differ based on the differentiation state and anatomical region from which the organoids are derived, but whether this affects the degree of SARS-CoV-2 infection is debated. A broader comparison of gene expression patterns and SARS-CoV-2 infection across organoids from different donors and culture conditions may help interpret the studies that have highlighted the extreme range of *ACE2* expression in intestinal tissues associated with demographic and clinical features of individuals, which could have consequences for susceptibility to both viral infections and inflammatory conditions [36–41].

Only small numbers of independent small intestinal and colonic organoid lines were compared to one another in previous studies examining SARS-CoV-2 infection of intestinal epithelial cells. Thus, the importance of the anatomical origin of organoids and other variables remains unclear. We established and differentiated 3D organoid lines from small intestinal and colonic biopsies procured from 12 and 13 donors, respectively, from healthy donors and patients with inflammatory bowel disease (IBD) of both sexes (S1 Table). The expression of *ACE2*, *TMPRSS2*, the enterocyte marker of differentiation *APOA1*, and representative interferon-stimulated genes (ISGs) *ISG15*, *OASL*, and *MX2* were significantly higher in small intestinal and colonic organoid lines cultured in 3D differentiation media (3DD) compared with those cultured in expansion media (3DE) that maintains organoids in an undifferentiated state (S1A, S1B, and S1D–S1G Fig). *TMPRSS4* expression was similar in both conditions (S1C Fig). Donor-to-donor variability in expression of these genes may reflect the inflammatory environment from which the stem cells were procured. However, 3DE organoids derived from IBD and non-IBD donors displayed comparable gene expression patterns except the decreased *OASL* expression in IBD donor-derived colonic 3DE organoids (S1H Fig). Intestinal organoids can be grown as differentiated monolayers to expose the apical side and facilitate viral entry. We found that the level of *ACE2*, *TMPRSS2*, *TMPRSS4*, *APOA1*, *ISG15*, *OASL*, and *MX2* expression in organoid-derived 2D monolayers correlated well with 3DD organoids generated from the same donor (Fig 1A–1D), suggesting that organoid lines retain their intrinsic gene expression patterns independent of these 2 culturing conditions. In addition, we found that monolayers exhibited the highest *ACE2* and *TMPRSS2* expression among the culture

forming unit; RNA-seq, RNA sequencing; RPMI, Roswell Park Memorial Institute; RT, room temperature; S, spike; SARS-CoV-2, Severe Acute Respiratory Syndrome Coronavirus 2; VSV-G, vesicular stomatitis virus G protein.

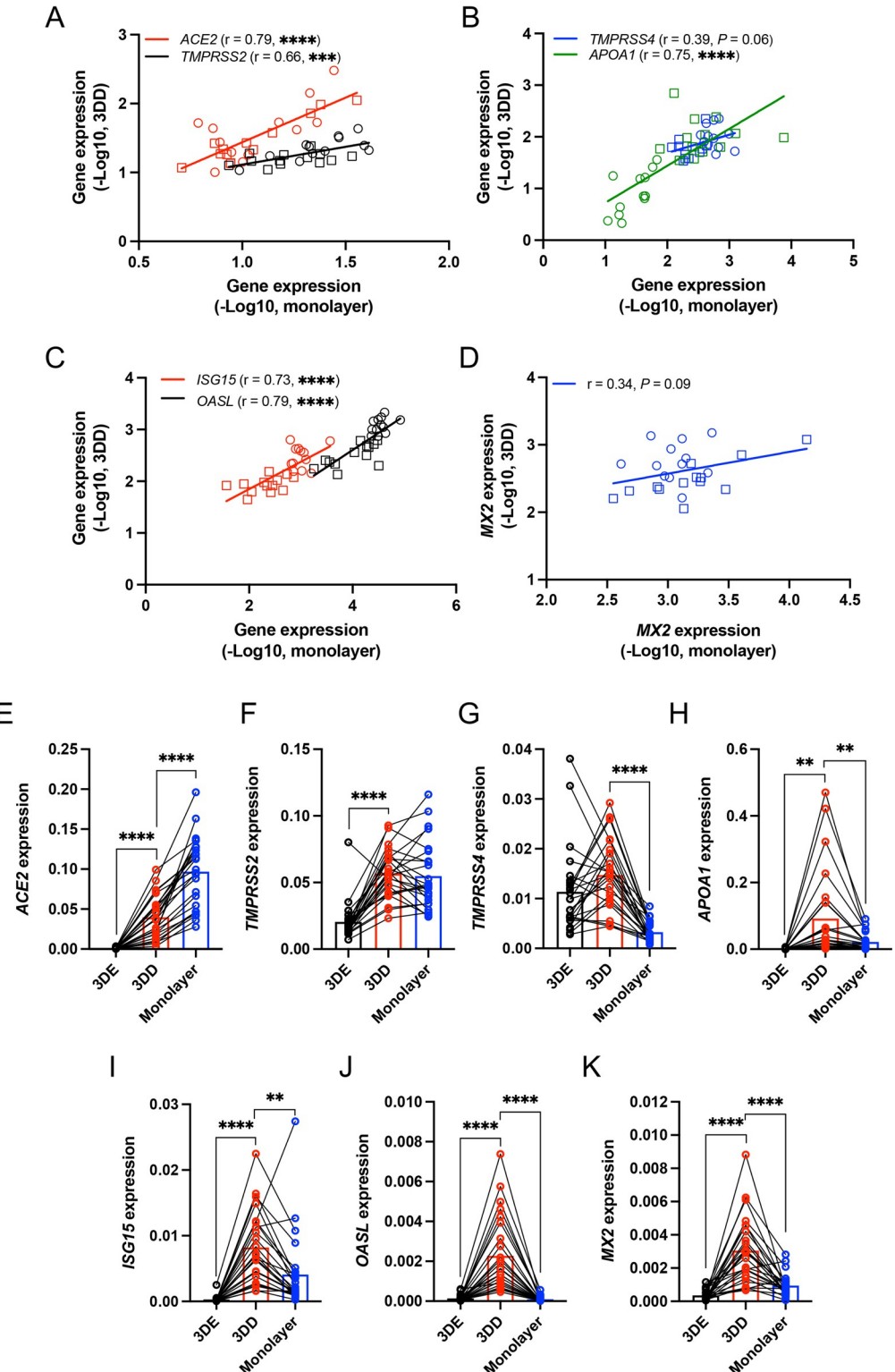

**Fig 1. Heterogeneous gene expression patterns in small intestinal and colonic organoids. (A–D)** Correlation analysis of *ACE2* and *TMPRSS2* (A), *TMPRSS4* and *APOA1* (B), *ISG15* and *OASL* (C), or *MX2* (D) expression in human intestinal (circle) and colonic (rectangle) organoids cultured as monolayers grown in differentiation media for 7 days with those cultured as human 3D organoids grown in differentiation media (3DD) for 7 days. **(E–K)** RT-PCR data comparing *ACE2* (E), *TMPRSS2* (F), *TMPRSS4* (G), *APOA1* (H), *ISG15* (I), *OASL* (J), and *MX2* (K) expression in

human 3D organoids grown with expansion media (3DE), 3DD, and monolayers grown in differentiation media for 7 days. Data points are mean of at least 2 technical replicates of individual organoid lines. Bars represent mean ± SEM, and at least 2 independent experiments were performed. Underlying data can be found in S1 Data. *P*, *P* value; r, Pearson correlation coefficient. **$P \leq 0.01$, ***$P \leq 0.001$, and ****$P \leq 0.0001$ by simple regression analysis in A–D and paired *t* test, 2 tailed in E–K. ACE2, angiotensin I converting enzyme 2; RT-PCR, reverse transcription PCR.

conditions we examined (Fig 1E and 1F), whereas 3DD organoids showed the highest *TMPRSS4*, *APOA1*, *ISG15*, *OASL*, and *MX2* expression (Fig 1G–1K). Therefore, we used the monolayer model for subsequent analyses.

We investigated whether *ACE2*, *TMPRSS2*, or *TMPRSS4* expression differed between small intestinal and colonic organoid–derived monolayers. Although *ACE2* expression did not differ, *TMPRSS*2 and *TMPRSS4* expression were higher, and *APOA1* was decreased in colonic monolayers (S1I–S1L Fig). Five pairs of the small intestinal and colonic organoid lines were generated from the same individual. We found that gene expression patterns were generally the same when comparing small intestinal and colonic monolayers from the same donor (S2A Fig). The lack of correlation between *ACE2* or *TMPRSS2* expression and *APOA1* expression suggested that heterogeneous *ACE2* and *TMPRSS2* levels were not an artifact caused by insufficient differentiation (S2B and S2C Fig). When we segregated the data based on disease status or sex, the only difference we observed was decreased *ACE2* and *APOA1* expression in colonic monolayers derived from IBD patients compared with non-IBD donors (S2D–S2F Fig). The expressions of *ISG15*, *OASL*, and *MX2* were also not correlated with disease status or sex (S2E and S2F Fig), although we note that *ISG15* and *OASL* transcripts were higher in colonic versus small intestinal monolayers (S3A and S3B Fig). These transcripts generally did not correlate with the age of participants (S2 Table).

*ACE2* expression differed by as much as 5.9-fold when comparing monolayers with the highest and lowest expression of this gene (Fig 2A). To test whether such differences lead to heterogeneity in viral infection, we infected monolayers with SARS-CoV-2 at a multiplicity of infection (MOI) of 4 for 72 hours. Remarkably, the amount of virus detected in the supernatant of culture media by plaque assay differed by as much as 423-fold (Fig 2B). Immunofluorescence microscopy analyses of ACE2 and SARS-CoV-2 nucleoprotein (NP) in representative monolayers confirmed these findings—SI1 and C1 (susceptible small intestinal and colonic monolayers, respectively, with high *ACE2* transcript levels) displayed higher degrees of ACE2 and NP staining compared with SI10 and C8 (resistant small intestinal and colonic monolayers, respectively) (Fig 2C and 2D). Indeed, SARS-CoV-2 infection correlated with *ACE2* and *TMPRSS2* expression, but not *TMPRSS4* and *APOA1* expression (Fig 2E–2H). We validated these findings based on relative *ACE2* transcript levels by enumerating absolute RNA copy numbers. The strong correlation of high copy numbers of *ACE2* ($2.9 \times 10^5$ to $1.7 \times 10^6$ transcripts/µg of RNA) with SARS-CoV-2 infection supported the relationship between susceptibility to infection and *ACE2* expression (S3D Fig). The amount of virus recovered from monolayers was comparable when the data were segregated by the tissue location, the disease status, or sex of the donors (S3E Fig). Similarly, virus production did not correlate with donor age (S3F Fig).

The association between susceptibility to SARS-CoV-2 infection and *ACE2* expression was clear in most cases, but there were outliers for the colonic monolayers. For instance, C7 and C8 have moderate to high levels of *ACE2* but low levels of virus production (Fig 2A and 2B). Thus, we investigated whether other factors may contribute to differential SARS-CoV-2 infectivity. Several polymorphisms are predicted to alter the stability of ACE2 or alter its affinity to the SARS-CoV-2 spike (S) protein [42–45]. However, the sequence of the *ACE2* coding region of C7 and C8 were identical to other donors (S1 Table). Differences in interferon responses

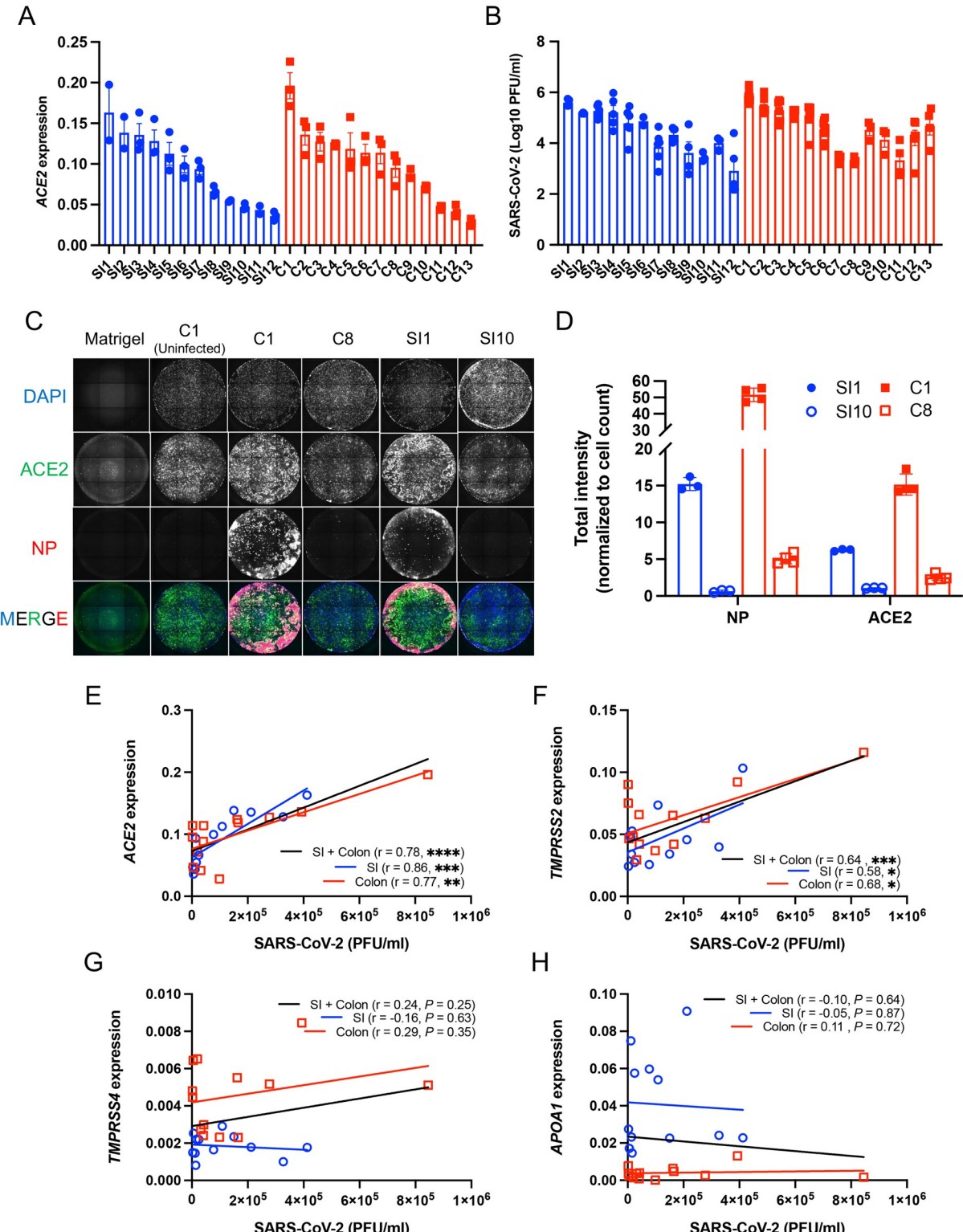

**Fig 2. Differential susceptibility of intestinal organoid–derived monolayers correlates with *ACE2* and *TMPRSS2* expression. (A)**
RT-PCR analysis of *ACE2* expression in small intestinal (SI1-SI12) and colonic (C1-C13) monolayers. **(B)** PFUs determined by virus titration on Vero E6 cells of supernatant from monolayers at 72 hours postinfection with SARS-CoV-2. **(C)** Representative immunofluorescence microscopy images showing co-staining of DAPI (blue), ACE2 (green), and SARS-CoV-2 NP (red) in SARS-CoV-2 infected SI1, SI10, C1, and C8 monolayer lines. An image of uninfected C1 is shown as a representative uninfected condition and a Matrigel-coated well without cells is

shown as a control for background fluorescence. **(D)** Total intensity of NP and ACE2 normalized with cell counts of SI1, SI10, C1, and C8. **(E–H)** Correlation of SARS-CoV-2 PFU with *ACE2* (E), *TMPRSS2* (F), *TMPRSS4* (G), or *APOA1* (H) expression among monolayers. Data points in A, B, and D are each technical replicate, and data points in E–H are the mean of at least 2 technical replicates of individual organoid lines. Bars represent mean ± SEM, and at least 2 independent experiments were performed. Underlying data can be found in S1 Data. *P*, *P* value; r, Pearson correlation coefficient. $^*P \leq 0.05$, $^{**}P \leq 0.01$, $^{***}P \leq 0.001$, and $^{****}P \leq 0.0001$ by simple regression analysis in E–H. ACE2, angiotensin I converting enzyme 2; NP, nucleoprotein; PFU, plaque-forming unit; RT-PCR, reverse transcription PCR; SARS-CoV-2, Severe Acute Respiratory Syndrome Coronavirus 2; SI, small intestine.

may also contribute to SARS-CoV-2 infectivity, where higher ISG levels are predicted to confer protection against viral infection [46–48]. Baseline *ISG15*, *OASL*, and *MX2* expression in C7 and C8 did not explain the lower virus production (S3A–S3C Fig), and we did not observe correlations between *ISG15*, *OASL*, or *MX2* and SARS-CoV-2 susceptibility (S3G–S3I Fig). Next, we examined ISG expression following stimulation with interferon beta (IFNβ) or interferon lambda 2 (IFNλ2). Although we observed varied levels of ISG induction, they were not associated with reduced viral infection (S4A Fig, S3 Table). Generally, we did not detect an association between the degree to which these ISGs were induced and properties of the donor tissue location, disease status, and age (S4A–S4C Fig, S3 Table). However, small intestinal monolayers from female donors displayed higher ISG expression than male donors following IFNβ or IFNλ2 stimulation (S4D and S4E Fig). *ACE2* and *TMPRSS2* expression were not altered by IFNβ or IFNλ2 (S5 Fig), indicating that *ACE2* and *TMPRSS2* are not ISGs in monolayers. We note that this limited survey of transcript level changes does not rule out a potential role for antiviral cytokines, and a comprehensive protein level analyses of immune mediators will be necessary to identify additional mechanisms of resistance.

Our results thus far are consistent with the possibility that *ACE2* gene expression is a key determinant of the degree to which the intestinal epithelium of an individual is susceptible to SARS-CoV-2 infection. As organoids are differentiated from primary stem cells and expanded in culture [49,50], it was unclear whether interdonor differences reflect ACE2 levels in the primary tissues. Therefore, we measured ACE2 protein by immunofluorescence microscopy in small intestinal and colonic sections from the same donors corresponding to individual lines of organoid-derived monolayers (Fig 3, S6 Fig). ACE2 staining was restricted to the epithelium and most intense along the apical brush border (villi in the small intestine and top of the crypts in the colon; Fig 3A and 3B, S6A and S6B Fig), consistent with our data and previous studies showing that *ACE2* expression is enriched in differentiated enterocytes [32,33]. Primary tissue specimens also displayed heterogeneous ACE2 protein levels (Fig 3C, S6 Fig). The ACE2 mean intensity was decreased in colonic sections of IBD patients, but did not differ when comparing tissue location or sex (Fig 3D). The mean intensity of ACE2 staining in intestinal tissue sections strongly correlated with *ACE2* transcript and SARS-CoV-2 levels in monolayers derived from the same individual, but not with age (Fig 3E). Therefore, organoid-derived monolayers retain the variable ACE2 levels from its original tissue.

To further investigate how susceptible and resistant organoids differ from each other, we selected the 3 colonic monolayer lines that each displayed high infection (HI; C1, C2, and C3) or low infection (LI; C8, C12, and C13) for RNA sequencing (RNA-seq) analysis. We validated the transcriptional and microscopy analyses (Fig 2A, 2C and 2D) by Western blot, which showed higher levels of ACE2 protein in HI compared with LI monolayers and comparable levels of TMPRSS2 protein (Fig 4A). We then infected the monolayers with SARS-CoV-2 for 24 and 72 hours (I24 and I72, respectively) and compared these samples with mock infected monolayers harvested at 0 and 72 hours (UI0 and UI72, respectively). SARS-CoV-2 continues to replicate in organoids after the initial 24 hours, likely due to the low proportion of cells that are initially infected [33]. We reasoned that sampling early and late time points may

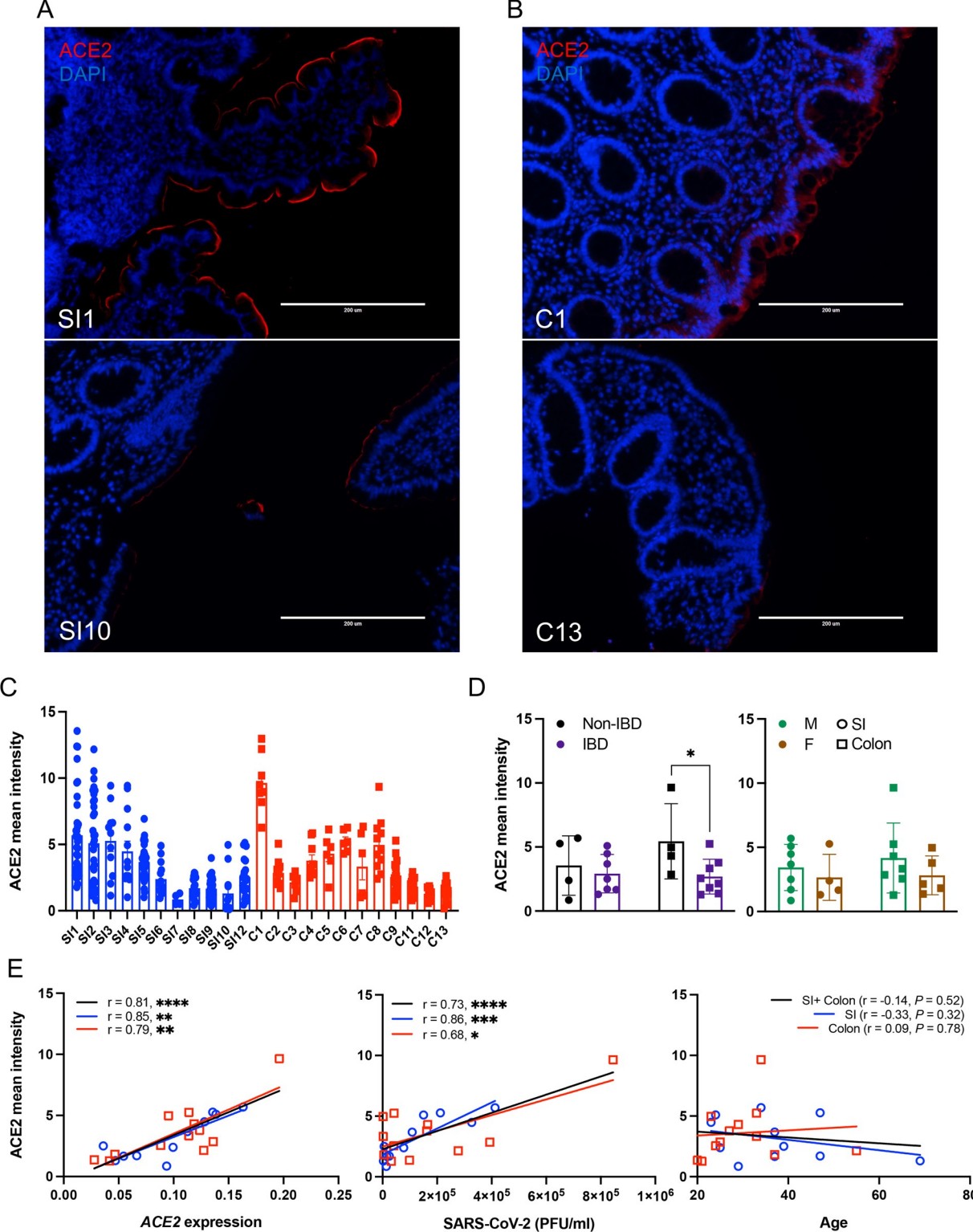

**Fig 3. Differential *ACE2* expression observed in individual intestinal organoid lines are conserved in primary tissue from the same donor.** (**A and B**) Representative ACE2 staining images in primary tissues of terminal ileum (A) and ascending colon (B) from which SI1, SI10, C1, and C13 organoids were established. (**C**) Mean intensity of ACE2 per area in each field of view from the primary tissues from which small intestinal and colonic organoids were established. (**D**) Mean intensity of ACE2 by disease (left) or sex (right). (**E**) Correlation of ACE2 mean intensity with *ACE2* expressions (left) and SARS-CoV-2 PFU (middle) among monolayers or donor age (right). Data points in C are the field of views, and

data points in D and E are mean of at least 2 technical replicates of individual organoid lines. Bars represent mean ± SEM, and at least 2 independent experiments were performed. Underlying data can be found in S1 Data. Scale bars: 200 μm. *P*, *P* value; r, Pearson correlation coefficient. *$P \leq 0.05$, **$P \leq 0.01$, ***$P \leq 0.001$, and ****$P \leq 0.0001$ by unpaired *t* test, 2 tailed in D and simple regression analysis in E. ACE2, angiotensin I converting enzyme 2; F, female; IBD, inflammatory bowel disease; M, male; PFU, plaque-forming unit; SARS-CoV-2, Severe Acute Respiratory Syndrome Coronavirus 2; SI, small intestine.

distinguish transcriptional changes that contribute to resistance and susceptibility to infection versus those that are a consequence. Because these 6 monolayer lines were prepared from independently thawed batches of frozen organoid stocks, we quantified virus and confirmed that higher amounts of SARS-CoV-2 were recovered from HI compared with LI monolayers at both time points (Fig 4B and 4C). Both HI and LI monolayers remained viable following SARS-CoV-2 infection (S7A Fig). Similar to our findings with IFN-stimulated monolayers (S5 Fig), *ACE2* and *TMPRSS2* transcripts in both HI and LI monolayers were stable during the course of infection (S7B and S7C Fig).

The number of transcripts displaying >2-fold changes (adjusted *P* value < 0.05) in 12 pairwise comparisons are summarized in S7D Fig. The conditions that displayed the most differences from one another were those comparing early time points to 72 hours postinfection. Uninfected samples at 0 and 72 hours displayed no differences, indicating that the transcriptome of uninfected HI and LI monolayers remained stable over time. This result increased our confidence that monolayers were fully differentiated at the onset of our experiments. Also, few genes displayed differential expression when comparing uninfected HI and LI monolayers. Principal component analysis (PCA) showed that infection at 72 hours separated samples on PC1 and susceptibility to infection (HI versus LI) separated samples on PC2 (S7E Fig). PCA also confirmed observations from the pairwise comparison indicating that uninfected monolayers and those infected for 24 hours were transcriptionally similar.

At the 72-hour time point, where the largest transcriptional changes occurred between conditions, we found that the majority of the differentially expressed genes (DEGs) when comparing uninfected and infected conditions (55 of 71) were common to LI and HI monolayers, while most DEGs (65 of 81) when comparing infected HI and LI monolayers were unique to this comparison (Fig 4D). Gene ontology analyses and ingenuity pathway analysis (IPA) showed that SARS-CoV-2 infection impacts antiviral pathways, especially those related to the interferon response, and that this signature was more pronounced in HI monolayers compared with LI monolayers following infection (Fig 4E–4G, S7F Fig). Indeed, DEGs related to the response to type I IFN (*SAMHD1*, *NLRC5*, *USP18*, *IFIT1*, *ZBP1*, *SHFL*, *STAT2*, *IRF7*, *SP100*, *MX1*, *OAS3*, *STAT1*, *ISG15*, and *OAS2*) and JAK-STAT (*STAT5A*, *STAT1*, *STAT2*, *CCL2*, and *NMI*) included common ISGs, and, although these were up-regulated in both HI and LI monolayers infected by SARS-CoV-2, they were induced to a higher degree in HI monolayers (S7G and S7H Fig). Because these ISGs are more highly expressed in HI monolayers and not detected at 24 hours, they are likely a response to the increased degree of infection. These results are consistent with other studies suggesting that the interferon response to SARS-CoV-2 is delayed [33,48,51–53]. Unexpectedly, multiple genes associated with pathways related to zinc and copper homeostasis were specifically up-regulated in LI compared to HI monolayers after 72 hours of infection (Fig 4E and 4H, S7I Fig). The increased expression of *MT1* genes encoding metallothioneins in LI monolayers was particularly striking and may be indicative of a stress response activated by viral perturbations in the epithelium [54,55]. Collectively, longitudinal transcriptome analyses identified robust yet late transcriptional changes induced by SARS-CoV-2, the magnitude of which corresponded to the levels of viral infection.

The transcriptome analysis did not provide additional insight into the difference in *ACE2* expression displayed by HI and LI monolayers. The transcription factors BRG1, FOXM1, and

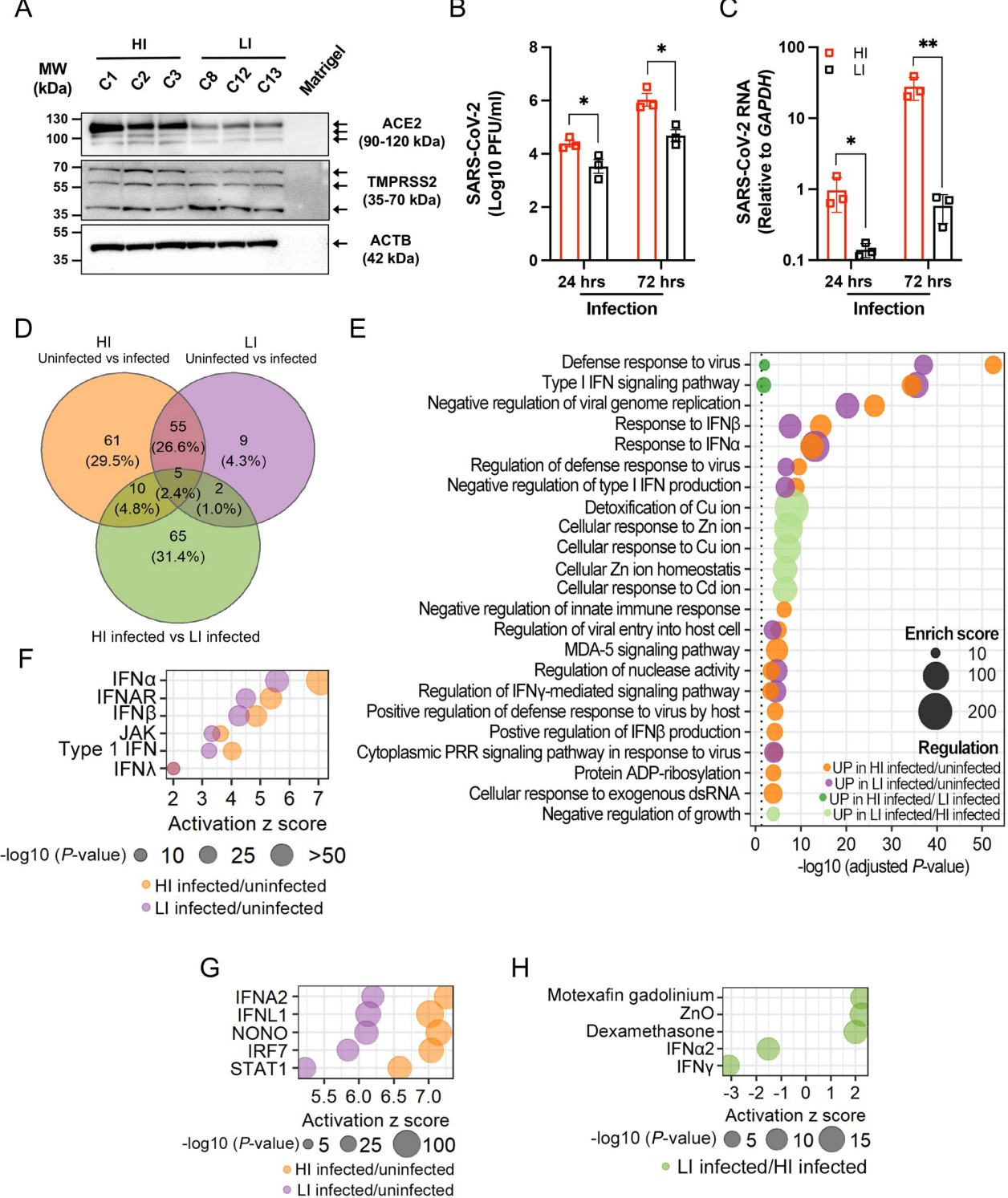

**Fig 4. Transcriptome analysis reveals a heighted and delayed interferon response to SARS-CoV-2 infection in susceptible organoid-derived monolayers. (A)** Western blot analysis of ACE2, TMPRSS2, and ACTB in high infection (HI; C1, C2, and C3) and low infection (LI; C8, C12, and C13) lines. Blots are representative of at least 3 independent repeats. **(B)** PFU determined by virus titration on Vero E6 cells of supernatant of HI and LI monolayers at 24 and 72 hours after infection with SARS-CoV-2. **(C)** RT-PCR analysis of SARS-CoV-2 expression in HI and LI monolayers upon 24 and 72 hours infection with SARS-CoV-2. **(D)** Venn diagram depicting the number and overlap of DEGs according to RNA-seq analysis (see S7 Fig) of

HI or LI monolayers infected with SARS-CoV-2 for 72 hours. (**E**) Highly enriched biological process GO terms for the DEGs in HI and LI infected with SARS-CoV-2. (**F and G**) IPA of the transcriptome of SARS-CoV-2–infected HI or LI for upstream regulators. Interferon-related genes (F) or top 5 molecules within the classes cytokine, transcription regulator, transmembrane receptor, ligand-dependent nuclear receptor, and others (G) commonly associated with HI infected/uninfected and LI infected/uninfected conditions. (**H**) DEGs of LI infected/HI infected condition were analyzed by IPA for upstream regulators. Top 5 upstream regulators related to zinc ion homeostasis and interferons for the LI. Data points in B and C represent the mean of at least 2 technical replicates of individual organoids lines. Bars represent mean ± SEM, and at least 2 independent experiments were performed. Underlying data can be found in S1 Data. $^*P \leq 0.05$ and $^{**}P \leq 0.01$ by unpaired $t$ test, 2 tailed in B and C. ACE2, angiotensin I converting enzyme 2; DEG, differentially expressed gene; GO, Gene Ontology; IFN, interferon; IFNα, interferon alpha; IFNAR, interferon alpha/beta receptor; IFNβ, interferon beta; IFNλ, interferon lambda; IPA, ingenuity pathway analysis; PFU, plaque-forming unit; PRR, pattern recognition receptor; RNA-seq, RNA sequencing; RT-PCR, reverse transcription PCR; SARS-CoV-2, Severe Acute Respiratory Syndrome Coronavirus 2.

FOXA2 mediate *ACE2* expression [56,57] but consistent with the RNA-seq results, HI and LI monolayers displayed comparable *BRG1*, *FOXM1*, and *FOXA2* expression (S8A–S8C Fig). However, we detected increased protein levels of FOXA2 in HI monolayers by Western blot, suggesting a probable mechanism for the high *ACE2* expression observed in these donors (S8D Fig).

During the preparation of this manuscript, a new variant of concern designated as Omicron emerged with multiple amino acid substitutions in the S protein. Thus, we examined the ability of the Omicron S protein to mediate entry into intestinal epithelial cells using SARS-CoV-2 S protein–pseudotyped lentiviral reporter viruses [58]. Vesicular stomatitis virus G protein (VSV-G) pseudotyped control virus displayed high infectivity of organoid-derived monolayers demonstrating feasibility of this approach (S9A Fig). Although Omicron S protein has been observed to have weaker or comparable binding affinity to ACE2 [59,60], Omicron S protein pseudotyped virus displayed 2.5- and 5-fold higher infection than Delta and D614G pseudo-types, respectively (S9A Fig), suggesting that Omicron exploits different or additional cell entry pathways to replicate in human intestinal organoids. Consistent with our observation, a recent study showed efficient entry of Omicron using the endosomal route [61]. D614G and Omicron S protein pseudotyped viruses showed 1.2- to 1.3-fold higher infection of HI mono-layers compared with LI monolayers, whereas the Delta S protein pseudotyped virus displayed comparable infectivity (S9B Fig). This marginal contribution of the differential *ACE2* expression to infection of these pseudotyped viruses suggests that other factors may be involved in SARS-CoV-2 susceptibility of intestinal epithelial cells. For example, *MT1* genes identified in our RNA-seq experiment (Fig 4E and 4H) are associated with resistance to hepatitis C virus and human cytomegalovirus, and zinc ion suppresses the SARS-CoV-2 replications by inhibiting its main proteases [62–65]. Although we caution against overinterpreting these results obtained with pseudotyped viruses, we believe that these preliminary results justify future studies using intestinal organoids and other donor-derived cell culture systems to examine differential susceptibility to intact viruses representing existing and future variants.

When taken together, our results show that human intestinal organoids reveal interindividual differences in responses to viral infection. Organoid-derived monolayers showed substantial differences in their susceptibility to SARS-CoV-2 infection, and ACE2 levels were the strongest correlate of susceptibility. Although transcriptome analysis identified many DEGs upon SARS-CoV-2 infection when comparing organoid lines, these differences were not apparent at 24 hours postinfection, a time point at which the degree of virus infection already diverged between resistant and susceptible monolayers. Therefore, these gene expression patterns are unlikely to account for differential susceptibility and instead, provide a glimpse as to how increased viral replication can affect properties of the intestinal epithelium. Although the presence of SARS-CoV-2 RNA in the gut has been associated with diarrhea in patients with COVID-19 [27], the consequence of intestinal epithelial infection remains largely unclear and an important area of investigation. Extensive experiments in animal models predict that

activation of viral RNA sensors trigger immune responses including ISGs that impact the intestinal barrier [66–76]. Coculturing organoids with leukocytes may help our understanding of the downstream consequences of epithelial infections [7]. Additionally, loss of microbiome diversity is associated with COVID-19 severity [77–80]. It would be important to determine whether the microbiome is involved in infection of the epithelium or represents an independent variable of disease outcome.

Finally, it is notable that organoids retained the differential ACE2 levels observed in intact primary tissue sections from the same donor. These results indicate that at least some transcriptional properties of the original intestinal epithelium that are individual specific are retained following ex vivo differentiation. Consistent with this theme, a recent study demonstrates that gastric organoids can be used to investigate age-dependent features of the stomach and yield insight into differential infection and interferon responses of children versus adults to SARS-CoV-2 [53]. If these findings are generalizable, then organoids can be a powerful platform to investigate interindividual differences in infectious disease susceptibility.

## Methods

### Human intestinal tissue specimen collection

Patients with and without IBD were recruited at outpatient colonoscopy performed for colon cancer screening, surveillance, or IBD activity assessment at New York University (NYU) Langone Health's Ambulatory Care Center, New York, under an NYU Grossman School of Medicine Institutional Review Board–approved study (Mucosal Immune Profiling in Patients with Inflammatory Bowel Disease; S12-01137). Approximately 6 pinch biopsies were obtained from the ascending colon of each patient using a 2.8-mm standard endoscopic biopsy forceps. The inflammation status of tissue was confirmed by endoscopic and histopathological examination. All pinch biopsies were collected in ice-cold complete Roswell Park Memorial Institute (RPMI) (RPMI 1640 medium supplemented with 10% fetal bovine serum (FBS), penicillin/ streptomycin/glutamine, and 50 μM 2-mercaptoethanol). Pinch biopsies were then transferred to freezing media (90% FBS + 10% DMSO) in cryogenic tubes and stored in liquid nitrogen.

### Culture of human small intestinal and colonic organoids

Human organoids were cultured as described previously [7,68]. Pinch biopsies were thawed in PBS and then incubated in Gentle Cell Dissociation Reagent (STEMCELL Technologies, Vancouver, BC, Canada) on ice for 30 minutes, followed by vigorous pipetting to isolate crypts. The crypts were embedded in 30 μl of Matrigel and cultured with Human IntestiCult Organoid Growth Medium (OGM) (Basal Medium and Organoid Supplement, STEMCELL Technologies) supplemented with 100 IU penicillin and 100 μg/ml streptomycin (Corning, Corning, NY, USA) and 125 μg/ml gentamicin (Thermo Fisher Scientific, Waltham, MA, USA), herein referred to as expansion medium. The culture medium was changed every 2 to 3 days. For passaging human organoids, 10 μM Y-27632 (Sigma-Aldrich, Burlington, MA, USA) were added for the first 2 days. For differentiation, the human organoids were cultured with 1:1 mix of Human IntestiCult OGM Basal Medium and Dulbecco's Modified Eagle Medium (DMEM)/F-12 (Thermo Fisher Scientific) in the presence of 100 IU penicillin and 100 μg/ml streptomycin, 125 μg/ml gentamicin, and 2 mM L-glutamine (Corning), herein referred to as differentiation medium. To generate organoid-derived monolayers, mature human small intestinal and colonic organoids grown with the expansion media were digested into single cells using TrypLE Express (Thermo Fisher Scientific) and seeded into Matrigel-coated 96-well culture plate (Corning) in Y-27632-supplemented expansion medium at 150,000 cells/well for the first 2 days. The culture media were changed every day. In experiments in which organoids were

treated with recombinant interferons, monolayers grown with the differentiation media for 7 days were stimulated with human IFNβ (100 or 300 IU/ml, R&D Systems, Minneapolis, MA, USA) or IFNλ2 (10 or 30 ng/ml, R&D Systems) for 12 hours.

## Transcript analysis

Total RNA was extracted from human organoids using RNeasy Mini Kit with DNase treatment (QIAGEN, Hilden, Germany), and synthesis of cDNA was conducted with High-Capacity cDNA Reverse Transcription Kit (Thermo Fisher Scientific) according to the manufacturer's protocol. RT-PCR was performed using SybrGreen (Roche, Basel, Switzerland) on a Roch480II Lightcycler using the following primers: *ACE2*, Fwd 5′-TCAAGGAGGCCGA GAAGTTC-3′ and Rev 5′-TTCCTGGGTCCGTTAGCATG-3′; *TMPRSS2*, Fwd 5′-ACCT GATCACACCAGCCATG-3′ and Rev 5′-CTTCGAAGTGACCAGAGGCC-3′; *TMPRSS4*, Fwd 5′-CCGATGTGTTCAACTGGAAG-3′ and Rev 5′-GAGAAAGTGAGTGGGAACTG-3′; *APOA1*, Fwd 5′-TGGATGTGCTCAAAGACAGC-3′ and Rev 5′-AGGCCCTCTGTCTCCT TTTC-3′; *ISG15*, Fwd 5′-GAGAGGCAGCGAACTCATCT-3′ and Rev 5′-CTTCAGCTCTG ACACCGACA-3′; *OASL*, Fwd 5′-AAAGAGAGGCCCATCATCC-3′ and Rev 5′-ATCTGGG TAACCCCTCTG C-3′; *MX2*, Fwd 5′-CAGCCACCACCAGGAAACA-3′ and Rev 5′-TTCTG CTCGTACTGGCTGTACAG-3′; BRG1, Fwd 5′-AGTGCTGCTGTTCTGCCAAAT-3′ and Rev 5′-GGCTCGTTGAAGGTTTTCAG-3′; FOXM1, Fwd 5′-GCAGGCTGCACTATCAAC AA-3′ and Rev 5′-TCGAAGGCTCCTCAACCTTA-3′; FOXA2, Fwd 5′-GGGAGCGGTGAA GATGGA-3′ and Rev 5′-TCATGTTGCTCACGGAGGAGTA-3′; *GAPDH*, Fwd 5′-GATGG GATTTCCATTGAT GACA-3′ and Rev 5′-CCACCCATGGCAAATTCC-3′; ACTB, Fwd 5′-CCCAGCCATGTACGTTGCTA-3′ and Rev 5′-TCACCGGAGTCCATCACGAT-3′; SARS-CoV-2 NP, Fwd 5′-ATGCTGCAATCGTGCTACAA-3′ and Rev 5′-GACTGCCGCC TCTGCTC-3′. The expression of the respective genes was normalized by geometric mean of *GAPDH* and *ACTB* expression with $\Delta\Delta C_T$ method [81]. Where indicated, the values were expressed as fold change from uninfected or untreated organoids. To determine the copy number of *ACE2* mRNA, a standard curve was constructed with the range of $1.4 \times 10^5$ to $9.4 \times 10^9$ molecules of hACE2 (Plasmid #1786, Addgene, Watertown, MA, USA) [82] in which *ACE2* transcripts showed optimal PCR efficiencies. The copy number of *ACE2* transcripts in the organoids was calculated from the linear regression of the standard curve and normalized with the RNA input.

## Virus infection and plaque assay

SARS-CoV-2 infection experiments were performed in the ABSL3 facility of NYU Grossman School of Medicine in accordance with its Biosafety Manual and Standard Operating Procedures. The organoid monolayers grown with the differentiation media for 7 days were infected with icSARS-CoV-2-mNG (isolate USA/WA/1/2020) obtained from the UTMB World Reference Center for Emerging Viruses and Arboviruses [83]. A working stock of SARS-CoV-2-mNG was generated by infecting a 90% to 95% confluent monolayer of Vero E6 cells (ATCC CRL-1586) for 48 hours at 37°C. Following incubation, the supernatant was collected, centrifuged at 1,200 rpm for 5 minutes, aliquoted, and stored at −80°C. Viral titers were quantified by plaque assay as described below. For the infection, organoid monolayers were infected with SARS-CoV-2-mNG at an MOI of 4 for 1 hour at 37°C. Following incubation, organoids were washed 4 times with PBS, and differentiation media was added for the indicated time. Total virus in the third or fourth wash was also quantified to ensure that excess virus was removed. Thus, virus quantified at the end of the experiment can be assessed as replicative particles rather than residual particles persisting in culture. Viral titers in the monolayer supernatants were quantified by plaque assay.

To quantify infectious virus by plaque assay, 10-fold serial dilutions of each sample were made in DMEM. Each dilution was added to a monolayer of Vero E6 cells for 1 hour at 37°C. Following incubation, DMEM supplemented with 2% FBS and 0.8% agarose was added and then incubated for 72 hours at 37°C. Cells were then fixed with 10% formalin, the agarose plug removed, and wells stained with crystal violet (10% crystal violet, 20% ethanol). Virus titers (plaque-forming unit [PFU]/ml) were determined by counting the number of plaques on the lowest countable dilution.

For thiazolyl blue tetrazolium bromide (MTT) reduction assay, staining with MTT was adapted from a previously described method [7]. Briefly, we added MTT (Sigma-Aldrich) into the monolayers to a final concentration of 500 μg/ml on 72 hours postinfection. After incubation for 2 hours at 37°C, 5% $CO_2$, the medium was discarded, and 20 μl of 2% SDS (Sigma-Aldrich) solution in water was added to solubilize the Matrigel for 2 hours. Then, 100 μl of DMSO (Thermo Fisher Scientific) was added for 1 hour to solubilize the reduced MTT, and Optical density was measured on a microplate absorbance reader (PerkinElmer, Waltham, MA, USA) at 562 nm. The specific organoid death (%) was calculated as MTT deduction (%) by normalizing to uninfected monolayers which were defined as 100% viable.

### *ACE2* sequencing

A 2,418-bp region containing the *ACE2* coding region was amplified from cDNA prepared from organoids (see above) by PCR using a pair of primers (Fwd 5′-ATGTCAAGCTCTTCC TGGCTCC-3′ and Rev 5′-CTAAAAGGAGGTCTGAACATCATCAGTG-3′). Amplicons were cloned into pCR2.1-TOPO (Invitrogen, Waltham, MA, USA). The plasmids were sequenced by Sanger sequencing from Psomagen (Rockville, MD, USA) using 4 primers: M13F, 5′-GTAAAACGACGGCCAGT-3′; M13R, 5′-CAGGAAACAGCTATGAC-3′; ACE2_600F, 5′- GGGGATTATTGGAGAGGAGACT-3′; ACE2_1800R, 5′-GTCGGTACTC CATCCCACA-3′.

### Immunofluorescence

ACE2 staining was performed as described previously [84]. Briefly, pinch biopsies of the terminal ileum and ascending colon were fixed in 10% formalin and embedded in paraffin blocks. Sections were cut to 5-μm thickness at the NYU Center for Biospecimen Research and Development and mounted on frosted glass slides. For deparaffinization, slides were baked at 70°C for 1.5 hours, followed by rehydration in descending concentration of ethanol (100%, 95%, 80%, 70%, ddH₂O twice; each step for 30 seconds). Heat-induced epitope retrieval was performed in a pressure cooker (Biocare Medical, Pacheco, CA, USA) using Dako Target Retrieval Solution, pH 9 (Dako Agilent, Santa Clara, CA, USA) at 97°C for 10 minutes and cooled down to 65°C. After further cooling to room temperature (RT) for 20 minutes, slides were washed for 10 minutes 3 times in TBS containing 0.1% Tween 20 (Sigma-Aldrich; TBS-T). Sections were blocked in 5% normal donkey serum (Sigma-Aldrich) in TBS-T at RT for 1 hour, followed by incubation with rabbit anti-ACE2 antibody (1:100, Abcam, Cambridge, England, ab15348) in the blocking solution at 4°C overnight. Sections were washed 3 times with TBS-T and stained with the Alexa Flour 555 conjugated with donkey anti-rabbit IgG (1:500, Thermo Fisher Scientific, A-31572) in PBS with 3% bovine serum albumin (BSA) (Thermo Fisher Scientific), 0.4% saponin, and 0.02% sodium azide at RT for 1 hour. Following this, sections were washed 3 times with TBS-T and mounted with ProLong Glass Antifade Mountant with NucBlue Stain (Thermo Fisher Scientific, P36918). Images were acquired using an EVOS FL Auto Cell Imaging System (Thermo Fisher Scientific) and then processed and quantified using ImageJ.

SARS-CoV-2–infected monolayers were fixed with 4% paraformaldehyde (Electron Microscopy Sciences, Hatfield, PA, USA) for 3 hours at RT. Following fixation, cells were washed 3 times with PBS then blocked and permeabilized in PBS with 0.1% Triton-X100 and 3% BSA for 30 minutes at RT. The permeabilized organoids were washed 3 times with PBS and incubated with mouse anti-SARS-CoV-2 N antibody (1:1,000, Prosci, Fort Collins, CO, USA 10–605) and rabbit anti-ACE2 antibody (1:500, Abcam, ab15348) diluted in PBS containing 3% BSA overnight at 4°C. The monolayers were washed 3 times with PBS and stained with Alexa Flour 647 conjugated with goat anti mouse IgG (1:2,000, Thermo Fisher Scientific, A32728) and Alexa Flour 594 conjugated with donkey anti rabbit IgG (1:1,000, Thermo Fisher Scientific, 21207) diluted in PBS with 3% BSA and DAPI for 1 hour at RT. The monolayers were then washed 3 times with PBS and imaged using the CellInsight CX7 High-content Microscope (Thermo Fisher Scientific).

## Immunoblotting

Organoids were processed for immunoblotting as previously described [76]. Briefly, monolayers were incubated in lysis buffer (20 mM Tris-HCl (pH 7.4), 150 mM NaCl, 1% Triton X-100, 10% glycerol, and 2x Halt Protease and Phosphatase Inhibitor Cocktail (Thermo Fisher Scientific)) on ice for 5 minutes and centrifuged at 14,000 rpm for 20 minutes. Samples were resolved on Bolt 4–12% Bis-Tris Plus Gels (Invitrogen), transferred onto polyvinylidene difluoride membranes, and blocked using 5% skim milk. The following antibodies were used for immunoblotting studies: mouse anti-β-actin (1:5,000, Sigma-Aldrich, AC-15), rabbit anti-ACE2 (1:1,000, Abcam, ab15348), mouse anti-TMPRSS2 (1:1,000, Santa Cruz Biotechnology, Dallas, TX, USA, sc-515727), rabbit anti-BRG1 (1:250, R&D Systems, MAB5738), mouse anti-FOXM1 (1:100, Santa Cruz Biotechnology, sc-271746), and rabbit anti-FOXA2 (1:200, Cell Signaling Technology, Danvers, MA, USA, 8186). Secondary antibodies (mouse anti-rabbit and goat-anti mouse, 211-032-171 and 115-035-174, respectively) were purchased from Jackson ImmunoResearch (West Grove, PA, USA).

## RNA deep sequencing and analysis

Monolayers were cultured in differentiation media for 7 days and then were infected with SARS-CoV-2 at an MOI of 4 for 24 hours or 72 hours before RNA extraction with 2 to 3 technical duplicates per line. CEL-seq2 was performed on 53 RNA samples of human organoids. Sequencing was performed on Illumina NovaSeq 6000 (Illumina, San Diego, CA, USA). RNA-seq results were processed using the R package "DESeq2" to obtain variance stabilized count reads, fold changes relative to specific condition, and statistical $P$ value. Analysis of the organoid transcriptome focused on DEGs, defined as the genes with an absolute fold change relative to specific condition >2 and an adjusted $P$ value < 0.05.

## Plasmids

SARS-CoV-2 S expression vectors have been previously described [58]. Briefly, the SARS-CoV-2 S expression vector pcCoV2.S.Δ19, *S* gene was amplified from pcCOV2.S (Wuhan-Hu-1/2019 SARS-CoV-2 isolate) [58] with a forward primer containing Kpn-I site and reverse primer that deleted the 19 carboxyl-terminal amino acids and contained Xho-I site. The amplicon was then cloned into the Kpn-I and Xho-I of pcDNA6 (Invitrogen). The D614G mutation was introduced by overlap extension PCR of the Δ19.*S* gene using internal primers overlapping the D614G mutation and cloned into pcDNA6. Mutations in Delta variant S were introduced by overlapping PCR overlapping each Delta mutations and cloned into pcDNA6. The Omicron S expression vector was chemically synthesized and cloned into pcDNA6 [85]. pcVSV-G, pLenti.GFP.Nluc, lentiviral packaging plasmids pMDL, and pRSV.Rev have been previously described [58].

### SARS-CoV-2 S lentiviral pseudotypes

SARS-CoV-2 variant S protein pseudotyped lentiviral stocks were generated by cotransfection of 293T cells ($4 \times 10^6$) with pMDL, pLenti.GFP-NLuc, pcCoV2.S.Δ19, and pRSV.Rev (4:3:4:1 mass ratio) by calcium phosphate coprecipitation as previously described [58]. VSV-G pseudotyped lentivirus was generated substituting the S protein vector for pcVSV-G. Two days post-transfection, supernatant was harvested and passed through a 0.45-μm filter and ultracentrifuged over a 20% sucrose cushion at 30,000 rpm for 90 minutes. The virus pellet was resuspended to 1/10 the initial volume in DMEM with 10% FBS, and virus titers were normalized by real-time PCR reverse transcriptase activity. Pseudotyped virus infectivity assay was done with HI and LI monolayers at an MOI of 0.2. After 72 hours of infection, luciferase activity was measured by Nano-Glo luciferase substrate (Promega, Madison, WI, USA) with an Envision 2103 microplate luminometer (PerkinElmer).

### Computational and statistical analysis

Gene ontology analysis was performed using the R package "clusterProfiler." PCA was performed using the R package "stats." Heatmaps were generated using the R package "pheatmap." Upstream regulators analysis was performed by uploading the DEGs to IPA software (QIAGEN). Statistical differences were determined as described in figure legend using either R or GraphPad Prism 9 software (La Jolla, CA, USA).

### Supporting information

**S1 Data. Excel spreadsheet containing, in separate sheets, the underlying numerical data for Figs 1A, 1B, 1C, 1D, 1E, 1F, 1G, 1H, 1I, 1J, 1K, 2A, 2B, 2D, 2E, 2F, 2G, 2H, 3C, 3D, 3E, 4B, 4C, 4E, 4F, 4G, and 4H, S1A, S1B, S1C, S1D, S1E, S1F, S1G, S1H, S1I, S1J, S1K, S1L, S2A, S2B, S2C, S2D, S2E, S2F, S3A, S3B, S3C, S3D, S3E, S3F, S3G, S3H, S3I, S4A, S4B, S4C, S4D, S4E, S5A, S5B, S6C, S7A, S7B, S7C, S7E, S7F, S7G, S7H, S7I, S8A, S8B, S8C, S9A, and S9B Figs, and S2 and S3 Tables.**
(XLSX)

**S1 Fig. Transcriptional analysis in SI and colonic organoids cultured in different media conditions. (A–G)** RT-PCR analysis of *ACE2* (A), *TMPRSS2* (B), *TMPRSS4* (C), *APOA1* (D), *ISG15* (E), *OASL* (F), and *MX2* (G) expression among SI or colonic 3D organoids grown in expansion media (3DE) or differentiation media (3DD) for 7 days. **(H)** RT-PCR data showing *ACE2*, *TMPRSS2*, *TMPRSS4*, *APOA*, *ISG15*, *OASL*, and *MX2* expression in 3DE organoids according to disease status. **(I–L)** RT-PCR analysis of *ACE2* (I), *TMPRSS2* (J), *TMPRSS4* (K), and *APOA1* (L) expression among SI and colonic monolayers grown in differentiation media for 7 days. Data points are mean of at least 2 technical replicates of individual organoid lines. Bars represent mean, and at least 2 independent experiments were performed. Underlying data can be found in S1 Data. P, P value. $^*P \leq 0.05$, $^{**}P \leq 0.01$, $^{***}P \leq 0.001$, and $^{****}P \leq 0.0001$ by paired *t* test, 2 tailed in A–G and unpaired *t* test, 2 tailed in H–L. ACE2, angiotensin I converting enzyme 2; ISG, interferon-stimulated gene; RT-PCR, reverse transcription PCR; SI, small intestine.
(TIF)

**S2 Fig. Transcript analysis of organoid-derived monolayers according to tissue location, disease, and sex of donors. (A)** RT-PCR analysis of *ACE2*, *TMPRSS2*, *TMPRSS4*, *APOA1*, *ISG15*, *OASL*, and *MX2* expression between matched donor-derived SI and colonic monolayers grown in differentiation media for 7 days. Data are displayed fold-change differences between the expression of the indicated genes in colonic monolayers over expression in SI

monolayers. **(B and C)** Correlation of *ACE2* (B) and *TMPRSS2* (C) expression with *APOA1* expression among monolayers grown in differentiation media. **(D)** RT-PCR analysis of *ACE2* expression among monolayers grown in differentiation media according to the disease status or sex of donors. **(E and F)** RT-PCR data depicting *ACE2*, *TMPRSS2*, *TMPRSS4*, *APOA1*, *ISG15*, *OASL*, and *MX2* expression of monolayers according to disease (E) or sex (F). Data points are mean of at least 2 technical replicates of individual organoid lines. Bars represent mean ± SEM, and at least 2 independent experiments were performed. Underlying data can be found in S1 Data. FC, fold change; r, Pearson correlation coefficient; *P*, *P* value. $^{*}P \leq 0.05$, $^{**}P \leq 0.01$, and $^{***}P \leq 0.001$ by paired *t* test, 2 tailed in A, simple regression analysis in B and C, and unpaired *t* test, 2 tailed in D–F. ACE2, angiotensin I converting enzyme 2; F, female; ISG, interferon-stimulated gene; M, male; RT-PCR, reverse transcription PCR; SI, small intestine.
(TIF)

**S3 Fig. Analysis of gene expression and viral replication in organoids by intestinal region, disease, sex, or age of donors. (A–C)** RT-PCR analysis of *ISG15* (A), *OASL* (B), and *MX2* (C) expression in monolayers. **(D)** Correlation of *ACE2* copy number with *ACE2* expression (left) or PFU of SARS-CoV-2 (right). **(E)** PFU of SARS-CoV-2 according to intestinal region (left), disease (middle), or sex (right) of donors. **(F–I)** Correlation of PFU of SARS-CoV-2 with age of donors (F) or *ISG15* (G), *OASL* (H), and *MX2* (I) expression. Data points are mean of at least 2 technical replicates of individual organoid lines. Bars represent mean ± SEM, and at least 2 independent experiments were performed. Underlying data can be found in S1 Data. r, Pearson correlation coefficient; *P*, *P* value. $^{*}P \leq 0.05$, $^{**}P \leq 0.01$, $^{***}P \leq 0.001$, and $^{****}P \leq 0.0001$ by unpaired *t* test, 2 tailed in A–C and E and simple regression analysis in D and F–I. ACE2, angiotensin I converting enzyme 2; F, female; M, male; PFU, plaque-forming unit; RT-PCR, reverse transcription PCR; SARS-CoV-2, Severe Acute Respiratory Syndrome Coronavirus 2.
(TIF)

**S4 Fig. Transcript analysis of IFNβ- or IFNλ2-stimulated organoids according to intestinal region, disease status, or sex of donors. (A–E)** RT-PCR data depicting fold change in *ISG15*, *OASL*, and *MX2* expression in SI and colonic monolayers stimulated with IFNβ (100 or 300 IU/ml) or IFNλ2 (10 or 30 ng/ml) for 12 hours according to intestinal region (A), disease status (B and C), or sex (D and E) of donors. Each value is normalized to nonstimulated organoid lines. Data points are mean of at least 2 technical replicates of individual organoid lines. Bars represent mean ± SEM, and at least 2 independent experiments were performed. Underlying data can be found in S1 Data. *P*, *P* value. $^{*}P \leq 0.05$ and $^{**}P \leq 0.01$ by unpaired *t* test, 2 tailed. F, female; IFNβ, interferon beta; IFNλ2, interferon lambda 2; ISG, interferon-stimulated gene; M, male; RT-PCR, reverse transcription PCR; SI, small intestine.
(TIF)

**S5 Fig. Stimulation with IFNβ or IFNλ2 does not alter *ACE2* and *TMPRSS2* expression in organoids. (A and B)** RT-PCR depicting *ACE2* (A) and *TMPRSS2* (B) expression in monolayers stimulated with IFNβ (100 or 300 IU/ml) or IFNλ2 (10 or 30 ng/ml) for 12 hours. Data points are mean of at least 2 technical replicates of individual organoid lines. Bars represent mean ± SEM, and at least 2 independent experiments were performed. Underlying data can be found in S1 Data. ACE2, angiotensin I converting enzyme 2; IFNβ, interferon beta; IFNλ2, interferon lambda 2; RT-PCR, reverse transcription PCR; SI, small intestine.
(TIF)

**S6 Fig. Representative images of ACE2 staining in primary intestinal tissue. (A and B)** Representative ACE2 staining images in primary tissues of terminal ileum from which SI (SI2-9 and 12, A) or colonic (C2-9, 11, and 12, B) organoid-derived monolayers were established. Images of remaining monolayers are included in main Fig 3. **(C)** Quantification of the total intensity of ACE2 staining and surface area in each visual field from the primary tissues from which the indicated organoid lines were established. Data points in C are the visual field. Bars in A and B: 200 μm. Bars in C represent mean ± SEM. Underlying data can be found in S1 Data. ACE2, angiotensin I converting enzyme 2; SI, small intestine.
(TIF)

**S7 Fig. Transcriptome and viability of SARS-CoV-2–infected organoids. (A)** Viability according to MTT reduction assay of SARS-CoV-2–infected high infection (HI; C1, C2, and C3) and low infection (LI; C8, C12, and C13) lines. **(B and C)** RT-PCR data depicting *ACE2* (B) and *TMPRSS2* (C) expression in HI and LI monolayers with or without SARS-CoV-2 infection. **(D)** Number of up- and down-regulated genes identified by RNA-seq analysis in the indicated pair-wise comparison of HI and LI monolayer lines at different time point postinfection. $n = 3$ organoid lines per group and at least 2 technical replicates per individual organoid line. **(E)** Unsupervised clustering based on expression of most variable genes by organoids lines and infection with SARS-CoV-2 at 24 and 72 hours. **(F)** Highly enriched molecular function GO terms for the DEGs in HI and LI infected with SARS-CoV-2 for 72 hours. **(G and H)** Heatmaps displaying normalized expression values of DEGs in HI infected/uninfected and LI infected/uninfected conditions (average fold-change >2 and adjusted $P$ value < 0.05) annotated in GO:0060338 and GO:0034340 (G) and GO0:07259 (H). **(I)** Heatmap displaying normalized expression values of DEGs in LI infected/HI infected conditions (average fold-change >2 and adjusted $P$ value < 0.05) annotated in GO:0006882. Data points in A–C are mean of at least 2 technical replicates of individual organoid lines. Bars represent mean ± SEM, and at least 2 independent experiments were performed. Underlying data can be found in S1 Data. UI0, uninfected 0 hour; UI72, uninfected 72 hours; I24, infected for 24 hours; I72, infected for 72 hours. ACE2, angiotensin I converting enzyme 2; DEG, differentially expressed gene; GO, Gene Ontology; MTT, thiazolyl blue tetrazolium bromide; RNA-seq, RNA sequencing; RT-PCR, reverse transcription PCR; SARS-CoV-2, Severe Acute Respiratory Syndrome Coronavirus 2.
(TIF)

**S8 Fig. Transcript and Western blot analyses of BRG1, FOXM1, and FOXA2. (A–C)** RT-PCR analysis of *BRG1* (A), *FOXM1* (B), and *FOXA2* (C) expression in HI and LI monolayers. **(D)** Western blot analysis of BRG1, FOXM1, FOXA2, and ACTB in HI and LI monolayers. Blots are representative of at least 2 independent repeats. Data points are mean of at least 2 technical replicates of individual organoid lines. Bars represent mean ± SEM, and 2 independent experiments were performed. Underlying data can be found in S1 Data. UI0, uninfected 0 hour; UI72, uninfected 72 hours; I24, infected for 24 hours; I72, infected for 72 hours. RT-PCR, reverse transcription PCR.
(TIF)

**S9 Fig. Infection of SARS-CoV-2 S protein–pseudotyped lentiviral reporter viruses. (A and B)** HI and LI monolayers were infected with VSV-G or SARS-CoV-2 D614G, Delta, or Omicron S protein pseudotyped viruses at an MOI of 0.2. At 72 hours postinfection, infectivity was measured by luciferase assay. Data points are mean of at least 2 technical replicates of individual organoid lines. Bars represent mean ± SEM, and 2 independent experiments were performed. Underlying data can be found in S1 Data. $P$, $P$ value. ****$P \le 0.0001$ by unpaired $t$

test, 2 tailed. MOI, multiplicity of infection; RLU, relative luminescence unit; S, spike; SARS-CoV-2, Severe Acute Respiratory Syndrome Coronavirus 2.
(TIF)

**S1 Table. Donor information for human endoscopic specimens.** CD, Crohn disease; F, female; IBD, inflammatory bowel disease; M, male; NA, not attempted; ND, not detected; SNP, single nucleotide polymorphism; UC, ulcerative colitis.
(PDF)

**S2 Table. Correlation between donor age and gene expression among the organoid monolayers grown in differentiation media for 7 days.** *P*, *P* value; r, Pearson correlation coefficient; SI, small intestine.
(PDF)

**S3 Table. Correlation of donor age or PFU of SARS-CoV-2 with ISG induction in organoids stimulated with IFNβ (100 or 300 IU/ml) or IFNλ2 (10 or 30 ng/ml) for 12 hours.** *P*, *P* value; r, Pearson correlation coefficient. $^*P \leq 0.05$ and $^{***}P \leq 0.001$ by simple regression analysis. IFNβ, interferon beta; IFNλ2, interferon lambda 2; PFU, plaque-forming unit; SARS-CoV-2, Severe Acute Respiratory Syndrome Coronavirus 2; SI, small intestine.
(PDF)

**S1 Raw Images. Raw Western blot images for Fig 4A and S8D Fig.**
(PDF)

## Author Contributions

**Conceptualization:** Kyung Ku Jang, Kenneth A. Stapleford, Ken Cadwell.

**Data curation:** Kyung Ku Jang, Simone Dallari, Kenneth A. Stapleford, Ken Cadwell.

**Formal analysis:** Kyung Ku Jang, Maria E. Kaczmarek, Ying-Han Chen.

**Funding acquisition:** Kenneth A. Stapleford, Ken Cadwell.

**Investigation:** Kyung Ku Jang, Maria E. Kaczmarek, Simone Dallari, Ying-Han Chen, Takuya Tada, Kenneth A. Stapleford.

**Methodology:** Kyung Ku Jang, Maria E. Kaczmarek, Simone Dallari, Ying-Han Chen, Takuya Tada, Kenneth A. Stapleford.

**Project administration:** Kenneth A. Stapleford, Ken Cadwell.

**Resources:** Kyung Ku Jang, Maria E. Kaczmarek, Simone Dallari, Ying-Han Chen, Jordan Axelrad, Nathaniel R. Landau.

**Supervision:** Kenneth A. Stapleford, Ken Cadwell.

**Validation:** Kyung Ku Jang, Maria E. Kaczmarek, Simone Dallari, Ying-Han Chen.

**Visualization:** Kyung Ku Jang, Maria E. Kaczmarek, Simone Dallari.

**Writing – original draft:** Kyung Ku Jang, Simone Dallari, Takuya Tada, Jordan Axelrad, Nathaniel R. Landau, Kenneth A. Stapleford, Ken Cadwell.

**Writing – review & editing:** Kyung Ku Jang, Maria E. Kaczmarek, Simone Dallari, Ying-Han Chen, Takuya Tada, Jordan Axelrad, Nathaniel R. Landau, Kenneth A. Stapleford, Ken Cadwell.

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
