## [Editor Report · Decision Letter 0]

25 Jul 2021

Dear Ken, 

Thank you for submitting your manuscript entitled "Intestinal organoids expose heterogeneity in SARS-CoV-2 susceptibility" for consideration as a Short Reports by PLOS Biology.

Your manuscript has now been evaluated by the PLOS Biology editorial staff [as well as by an academic editor with relevant expertise] and I am writing to let you know that we would like to send your submission out for external peer review.

Please re-submit your manuscript within two working days, i.e. by Jul 27 2021 11:59PM.

Kind regards,

Paula

---

Paula Jauregui, PhD

Associate Editor

PLOS Biology

---

## [Decision Letter · Decision Letter 1]

13 Sep 2021

Dear Dr. Cadwell,

Thank you for submitting your manuscript "Intestinal organoids expose heterogeneity in SARS-CoV-2 susceptibility" for consideration as a Short Reports at PLOS Biology. Your manuscript has been evaluated by the PLOS Biology editors, an Academic Editor with relevant expertise, and by several independent reviewers.

In particular, reviewer #1 would like you to clarify the conceptual advance of this work and the rationale to compare gene expression between SI and colonic organoids, wants you to do more in depth analyses to understand basal differences from IBD and non-IBD patients and other pathways that may correlate with SARS-cov2 infection, and asks for justification for the combination of SI and colon in one bar for statistical comparisons, about the time point of 72 h after infection, and the significance of the transcriptional analyses. Reviewer #2 has issues with the use of monolayers and the term "organoids", thinks that transcriptome sequencing of the organoids will help clarify the issues about ACE2 expression levels, thinks that the messages throughout the manuscript are confusing, thinks that the correlation in fig 2F is not strong, is not convinced by the ACE2 staining in the imaging, and would like you to add validations from an additional number of individuals. Reviewer #3 thinks that you should mention the possible role the microbiome might play in SARS2 infection in the GI tract, and thinks that additional measures of differential responses that rely on IFN protein production (e.g., ELISA) would enhance the rigor and impact of this aspect of the manuscript.

In light of the reviews (below), we will not be able to accept the current version of the manuscript, but we would welcome re-submission of a much-revised version that takes into account the reviewers' comments. We cannot make any decision about publication until we have seen the revised manuscript and your response to the reviewers' comments. Your revised manuscript is also likely to be sent for further evaluation by the reviewers.

We expect to receive your revised manuscript within 3 months. 

**IMPORTANT - SUBMITTING YOUR REVISION**

*Re-submission Checklist*

*Published Peer Review*

*PLOS Data Policy*

*Blot and Gel Data Policy*

Sincerely,

Paula

---

Paula Jauregui, PhD

Associate Editor

PLOS Biology

REVIEWS:

Reviewer #1: Inflammatory bowel disease.

Reviewer #2: Organoids.

Reviewer #3: Host-virus interactions.

Reviewer #1: This is a well written manuscript that examines patient-derived intestinal organoids. The main conclusion is that Ace2 mRNA expression levels correlate with SARS-CoV-2 burden in human intestinal organoids that are exposed to the pathogen. In addition, the authors indicate that human intestinal organoids are useful models to study infection. While examining SARS-CoV-2 with primary human cells is certainly of broad interest, the described findings seem to lack the novel conceptual advance needed to warrant publication as a Short Report.

Some specific concerns:

1. The authors acknowledge that studies have shown that human intestinal organoids support SARS-cov2 infection and there is known variability in Ace2 expression. Their data here confirms the correlation, however does not go further or test function. Thus, the conceptual advance of this work is not clear. 

2. Not clear on the rationale to compare gene expression between SI and colonic organoids (Fig 1F), especially since the SI and colonic organoids represent different patients and some are controls while others are derived from patients with IBD. It is not surprising that there would be extensive variability given these factors. 

3. Given the large initial differences in cells obtained from IBD and non-IBD patients, more in depth analyses should be included to understand basal differences in these conditions and other pathways that may correlate with SARS-cov2 infection. Figure 3D combines SI and colon in one bar for statistical comparisons. This should be justified.

4. What is the significance of 72hours post infection for organoids? It is not clear why this is an appropriate time point to examine infection in vitro. Is there variability in other factors- cell death?

5. In the same regard, what is the significance of the transcriptional analyses included in Figure 4. It seems expected that increased viral load would result in an increased magnitude of differences in viral-induced genes.

Reviewer #2: In the manuscript entitled "Intestinal organoids expose heterogeneity in SARS-CoV-2 susceptibility", Jang and colleagues seek the inter-individual variability of ACE2 RNA and protein expression and its relationship with SARS-CoV-2 infection levels. For the study, the authors establish patient-derived organoids, followed by in vitro infection with SARS-CoV-2. 

Although many figure panels are shown, the key message is straightforward. (1) Gut organoids show variability in the ACE2 expression level (2) The differential ACE2 expression in organoids reflects ACE2 expression level in the primary tissue, and (3) ACE2 expression levels of organoids show a good correlation with their SARS-CoV-2 susceptibility. Although they are interesting observations, these are not very surprising given the fact that patient-derived organoids are likely to have cellular characteristics of the specific individuals and the fact that ACE2 is the most crucial entry protein for SARS-CoV-2.

Before its publication, the reviewer thinks that a few key observations should be clarified further.

Major comments

(1) In Fig 1, the authors explored organoids in three different environments, i.e., (1) expansion media (3DE), (2) differentiation media (3DD), and (3) monolayer. Throughout the study, the authors used (3) monolayer conditions because the condition up-regulated ACE2 levels. However, by definition, organoids are 'three-dimensional' culture models developed from a single-stem cell, and the cell cultures in the monolayer condition may not be classified as 'typical organoids', but just patient-derived primary cellular models. In such conditions, can we generalize that their findings are from "organoids"?

(2) In Fig 1, the authors have shown that organoids in 3DD, 3DE, and monolayer show good correlations in ACE2 RNA expression levels. However, because ACE2 expression levels (and also other genes) were measured by qPCR, (1) the ACE2 expression levels are normalized by a single housekeeping gene (GAPDH), and (2) their absolute gene expression levels cannot be understood. Transcriptome sequencing of the organoids will help clarify the issues.

(3) Because the authors simultaneously compare (the colon vs the small intestine) AND (individual variations within the tissue), their messages throughout the manuscript (figures) are a bit confusing.

(4) Figure 2F is one of the most important figures in the manuscript. However, excluding 1-2 outliers, the correlation does not seem to be very strong, particularly in the colon. 

(5) The correlation between ACE2 protein levels in the primary tissues and ACE2 RNA-expression levels in the organoids is another critical part of the manuscript. The relationship was correlated in Figs 3A, 3B and 3E. Although I am not an expert in imaging, the ACE2 staining (Figs 3A, 3B) does not seem to be very convincing. Is there no ACE2 protein expression in SI7 and SI10?

(6) In line with (5), the correlation of the central panel of Fig 3E does not seem to be very strong when we remove the single-outlier with the highest ACE2 protein expression and SARS-CoV-2 infection levels. To clarify it, validations from an additional number of individuals should be necessary. 

Reviewer #3: This is an interesting Short Report that utilizes a biobank of human-derived intestinal organoids to investigate differences in SARS-CoV2 replication in the GI tract. The authors collected small intestine and colon biopsies from 8-10 donors, cultured organoids from these tissues under differentiated and undifferentiated states and in three-dimensions or two-dimensions, and then profiled the susceptibility of these parallel models to SARS2 infection as well as defining the host response to this infection. The authors conclude that expression of ACE2 is a primary determinant of SARS2 tropism between small intestine- and colon-derived organoids and that differential innate immune responses to infection correlate with differences in replication levels between these two systems. Although other groups have used GI-derived organoids for SARS2 studies, a clear strength of the current manuscript is the breadth of human biopsies and comparisons between differentiation states and growth conditions. The manuscript is very well-written and the conclusions are supported by rigorous parallel approaches. I have a few relatively minor points that could improve the clarity and/or conclusions of the study, which I have detailed below.

1. It would seem important for the authors to mention the possible role the microbiome might play in SARS2 infection in the GI tract, which might have a significant impact on human-to-human variability and would occur independent of ACE2 (or other host factor) expression.

2. While the differential expression analysis is compelling, additional measures of differential responses that rely on IFN protein production (e.g., ELISA) would enhance the rigor and impact of this aspect of the manuscript. If the authors are unable to perform these studies, they might wish to make clearer in the manuscript that transcript levels of IFNs are not always direct correlated to protein levels, especially between IFN types.

---

## [Editor Report · Decision Letter 2]

16 Feb 2022

Dear Ken,

Thank you for submitting your revised Short Reports entitled "Variable susceptibility of intestinal organoid-derived monolayers to SARS-CoV-2 infection" for publication in PLOS Biology. 

I have now discussed your new version with the Academic Editor and am pleased to let you know that we will probably accept this manuscript for publication, provided you satisfactorily address the following data and other policy-related requests:

1) Data:

1.1) Thank you for providing an accession number for your gene expression data (https://www.ncbi.nlm.nih.gov/geo/query/acc.cgi?acc=GSE179949). We note that the record is currently private and scheduled to be released on Jul 31, 2022 . Please ensure that this record is made publicly available now. 

1.2) Please ensure that figure legends in your manuscript include information on where the underlying data can be found (e.g. ‘The data underlying this Figure may be found in S1 Data’), and ensure your supplemental data file/s also has a legend.

1.3) Please ensure that your Data Statement in the submission system accurately describes where your data can be found.

2) Blot and gel reporting requirements: thank you for providing the western blots corresponding to Fig 4 A and Supplementary Fig 8 D. Please ensure that you provide the original, uncropped version of these blots without the red lines. Our guidelines for how to prepare and upload these data can be found at https://journals.plos.org/plosbiology/s/figures#loc-blot-and-gel-reporting-requirements.

3) Blurb: Please also provide a blurb which (if accepted) will be included in our weekly and monthly Electronic Table of Contents, sent out to readers of PLOS Biology, and may be used to promote your article in social media. The blurb should be about 30-40 words long and is subject to editorial changes. It should, without exaggeration, entice people to read your manuscript. It should not be redundant with the title and should not contain acronyms or abbreviations. For examples, view our author guidelines: https://journals.plos.org/plosbiology/s/revising-your-manuscript#loc-blurb

4) Related reference: we note that a related study has been published in November 2021, PMID: 34785679. Please ensure that this study is cited and discussed in the discussion section of your manuscript, as appropriate. 

We expect to receive your revised manuscript within two weeks. 

*Published Peer Review History*

*Early Version*

Sincerely,

Dario

Dario Ummarino, PhD

Senior Editor

PLOS Biology

dummarino@plos.org

---

## [Editor Report · Decision Letter 3]

4 Mar 2022

Dear Ken,

On behalf of my colleagues and the Academic Editor, Bon-Kyoung Koo, I am pleased to say that we can in principle accept your Short Reports "Variable susceptibility of intestinal organoid-derived monolayers to SARS-CoV-2 infection" for publication in PLOS Biology, provided you address any remaining formatting and reporting issues. These will be detailed in an email that will follow this letter and that you will usually receive within 2-3 business days, during which time no action is required from you. Please note that we will not be able to formally accept your manuscript and schedule it for publication until you have any requested changes.

PRESS

Sincerely, 

Dario

Dario Ummarino, PhD 

Senior Editor 

PLOS Biology

dummarino@plos.org